# Study of the Dynamic Model and Vibration Performance of Pot-Shaped Metal Rubber

**DOI:** 10.3390/ma15175878

**Published:** 2022-08-25

**Authors:** Xiaoyuan Zheng, Wei Wang, Yiwan Wu, Hongbai Bai

**Affiliations:** 1College of Mechanical Engineering and Automation, Fuzhou University, Fuzhou 350116, China; 2Metal Rubber Engineering Research Center, Fuzhou University, Fuzhou 350116, China

**Keywords:** metal rubber, dynamic model, nonlinear characteristics, parameter identification

## Abstract

Metal rubber has been extensively used in recent years due to its several unique properties, especially in adverse environments. Although many experimental studies have been conducted, theoretical research on metal rubber is still in its infancy. In this work, a dynamic model for the nonlinear characteristics of pot-shaped metal rubber is established on the basis of the asymmetric dynamic model of the wire rope shock absorber and the trace method model. In addition, the corresponding parameters in the model are identified based on the parameter-separation method. The theoretical hysteresis loop obtained using the model and the measured hysteresis loop agree with each other. The results show that the asymmetric dynamic model can better describe the asymmetric dynamic characteristics of pot-shaped metal rubber. Furthermore, a pot-shaped metal rubber vibration reduction system is built to further verify the correctness of the model. This study provides an experimental reference and a theoretical basis for the practical application of pot-shaped metal rubber in the field of three-dimensional vibration reduction.

## 1. Introduction

Metal rubber is a porous damping material generally made of a series of coiled metal wires with various sections embedded and hooked to achieve a spatial network structure [1,2,3,4,5]. Metal rubber has both the stiffness characteristics of pure metal and the damping characteristics of polymer rubber. It can adapt to harsh working conditions and has a long service life. Therefore, it is widely used in the aerospace industry, armored vehicles, and ship pipelines [6,7,8]. However, due to the disorder of the three-dimensional porous structure of metal rubber, research on the mechanical properties of metal rubber is limited. Iwan et al. [9] proposed a double-broken-line model, which represents the restoring force as the superposition of an elastic force and a hysteretic damping force. This method can accurately model the dry friction situation. However, metal rubber materials with strong nonlinear characteristics cannot be well described by this model because the damping force is treated only as a linear force. Zhang et al. [10] proposed a semi-constitutive dynamic model that involves nonlinear elastic stiffness, nonlinear viscous damping, and bilinear hysteresis Coulomb damping. Bouc et al. [11] proposed a first-order nonlinear equation model, which nonlinearizes the restoring force and predicts the evolution of the restoring force; this model is widely used in the random vibration response analysis of hysteretic systems. Yan et al. [12] used the double-discount model to describe metal rubber vibration isolators, and the energy method and the least square method were employed to identify the parameters. The identification results were found to be in good agreement with the experimental results. Li et al. [13] designed a metal rubber damper and established the system dynamics model based on the double-broken-line model. The niche genetic algorithm was used to identify the parameters, and the results showed that this method is simple, feasible, and accurate. Xue et al. [14] established a macroscopic mechanical model that can describe the nonlinear characteristics of metal rubber and verified the correctness of the model using specimens with different densities. Li et al. [15] conducted dynamic mechanical tests on braided-grooved metal rubber, and established a nonlinear dynamic model. The variation in the model parameters with displacement and the effects of the amplitude and excitation frequency on the dynamic characteristics of the vibration isolation system were analyzed. Considering the effect of temperature on the mechanical properties of metal rubber, Zi et al. [16] constructed a high-temperature dynamic model of a metal rubber coated damping structure of a pipeline using the equivalent linearization method.

At present, the establishment of dynamic models of metal rubber materials is mostly based on a symmetric model with a regular shape [17,18]. However, there are few studies regarding asymmetric dynamic models of metal rubber. In addition, the most regular shape of the metal rubbers is cylindrical, annular, and square, and there is little research on special-shaped metal rubbers. Pot-shaped metal rubber has the function of three-way vibration reduction and negative stiffness characteristics. In this study, we address the difficult problem of the nonlinear dynamic modeling of pot-shaped metal rubber and based on the analysis and study of the asymmetric model of the wire rope shock absorber combined with the trace model, a dynamic model suitable for describing the nonlinear characteristics of metal rubber is established. The corresponding parameters in the model are identified based on the parameter-separation identification method, and the experimental results are in excellent agreement with the theoretical results.

## 2. Asymmetric-Hysteresis Dynamic Model of Pot-Shaped Metal Rubber

At present, the research on the model of symmetric hysteresis loops mainly includes the trace method model and the double-broken-line model. However, the double-broken-line model involves a double-broken-line restoring force, which makes it difficult to identify the parameters. Ni et al. [19] proposed the following asymmetric hysteresis model:(1)$$F\left(t\right)=F_{2a}\left(t\right)[z\left(t\right)+F_{1a}(t)]$$
(2)$$F_{1a}\left(t\right)=k_{1}x\left(t\right)+k_{2}\mathrm{sgn}\left(x\right)x^{2}\left(t\right)+k_{3}x^{3}\left(t\right)$$
(3)$$F_{2a}\left(t\right)=b^{cx\left(t\right)}$$
(4)$$\dot{z}\left(t\right)=\dot{x}\left(t\right)\left\{\alpha -\left[\gamma +\beta \mathrm{sgn}\left(\dot{x}\left(t\right)\right)\right]\right.\mathrm{sgn}\left(\dot{x}\left(t\right)\right)\cdot \left.{\left|z\right|}^{n}\right\}$$
where *x* is the deformation; *t* is the time; *F* is the shock absorber load; *k*_1_, *k*_2_, *k*_3_, *b*, *c*, *α*, *β*, *γ*, and *n* are the model parameter; *F*1*a*(*t*) and *F*2*a*(*t*) are the structural modulation terms; $\dot{z}(t)$ is the Bouc–Wen hysteresis model [20].

However, Wu et al. [21] found that the parameter identification obtained using Equations (1)–(4) is not ideal. The calculation process can easily give rise to an ill-conditioned matrix; thus, an asymmetric model is proposed. It can be seen that the *z*(*t*) + *F*_1*a*_(*t*) sum in Equation (1) is a symmetric curve about the origin, and the role of *F*_2*a*_(*t*) is to transform the symmetric curve into an asymmetric one centered at the origin. However, the origin of the hysteresis loop obtained when using the model described by Ni [19] is not at the center of the curve. Therefore, in order to accurately describe the test curve, a constant term *h* should be added to Equation (1) so that the origin of the hysteresis loop is at the center of the curve. In addition, the simple harmonic motion can be expressed as:(5)$$x=A\sin\left(\omega t+\varphi \right)$$
where, *A* denotes the displacement amplitude, ω denotes the loading period, and φ represents the initial loading phase.

The trace model has a simple structure and can describe a variety of damping components, so it is widely used in the description of hysteresis loops in different engineering problems. According to the trace method, the restoring force comprises a nonlinear elastic restoring force and a hysteretic nonlinear damping force. The *F*_1*a*_(*t*) and *z*(*t*) can be regarded as the nonlinear elastic restoring force *F_k_* and the nonlinear damping force *F_c_*, respectively. The damping component is simplified to an equivalent viscous damping and expressed as a power function of velocity. The improved model can be expressed as follows:(6)$$F\left\{x\left(t\right),\dot{x}\left(t\right),t\right\}=F_{2a}\left(x\left(t\right)\right)[F_{c}\left(\dot{x}\left(t\right)\right)+F_{1a}\left(x\left(t\right)\right)]+h$$
(7)$$F_{1a}\left(x\left(t\right)\right)=k_{1}x\left(t\right)+k_{3}x^{3}\left(t\right)+k_{5}x^{5}\left(t\right)$$
(8)$$F_{2a}\left(x\left(t\right)\right)=e^{bx\left(t\right)}$$
(9)$$F_{c}\left(\dot{x}(t)\right)=c\mathrm{sgn}\left(\dot{x}\left(t\right)\right){\left|\dot{x}\left(t\right)\right|}^{\alpha }$$
where *h* represents the offset of the hysteresis loop center; *b* is the asymmetry parameter of the hysteresis loop; *k*_1_, *k*_3_, and *k*_5_ are the first-order, third-order, and fifth-order stiffness coefficients, respectively; $x\left(t\right)$ and $\dot{x}\left(t\right)$ are deformation and velocity, respectively; *c* is the damping coefficient; *α* is the damping component factor; $F_{1a}\left(x\left(t\right)\right)$ and $F_{2a}\left(x\left(t\right)\right)$ are the structural modulation terms.

Equations (6)–(9) is the asymmetric dynamic model of metal rubber components. The parameters of the model can be theoretically identified using the least square method. However, the model contains exponential terms, which may lead to an ill-conditioned matrix of the recognition results and increase the difficulty of the parameter identification. Therefore, the parameter-separation identification method is used for parameter identification.

## 3. Parameter Identification Method of the Model

### 3.1. Identification of the Parameters h and b

A typical asymmetric hysteresis loop of pot-shaped metal rubber is shown in Figure 1a. It can be seen from the figure that the origin of the coordinate system is not at the center of the hysteresis loop. Assuming that the coordinates of the points on the hysteresis loop are $\left(x_{i},F_{i}\right)$ and the restoring forces of points C and D on the upper and lower half branches are *F_C_*_0_ and *F_D_*_0_, then $h=\left(F_{C0}+F_{D0}\right)/2$.

Next, we carry out a coordinate transformation of the original hysteresis loop data, namely $x_{i}{}^{\prime }=x_{i},F_{i}{}^{\prime }=F_{i}-h$, and substitute these expressions into Equation (6) to obtain the following equation:(10)$$F^{\prime }\left(x\right)=e^{bx^{\prime }}[F_{c}\left(x^{\prime }\right)+F_{1a}(x^{\prime })]=e^{bx^{\prime }}F_{d}\left(x^{\prime }\right)$$

The hysteresis loop after transformation is shown in Figure 1b, where the origin of the coordinate system is at the center of the hysteresis loop.

### 3.2. Identification of the Parameter b

By substituting the transformed curve points $A\left({x^{\prime }}_{A},{F^{\prime }}_{A}\right),\;B\left({x^{\prime }}_{B},{F^{\prime }}_{B}\right)$ in Figure 1b into Equation (10), one obtains:(11)$$\left\{\begin{array}{l} F_{A}\left({x^{\prime }}_{A}\right)=e^{b{x^{\prime }}_{A}}F_{d}\left({x^{\prime }}_{A}\right)\\ F_{B}\left({x^{\prime }}_{B}\right)=e^{b{x^{\prime }}_{B}}F_{d}\left({x^{\prime }}_{B}\right) \end{array}\right.$$
where $F_{d}$ is an odd function symmetric about the origin, and ${x^{\prime }}_{A}=-{x^{\prime }}_{B}=x_{M}$. Thus, the following relationship holds:(12)$$F_{d}\left({x^{\prime }}_{A}\right)=-F_{d}\left({x^{\prime }}_{B}\right)$$
where $x_{M}$ is the maximum deformation. By substituting Equation (12) into Equation (11), one obtains:(13)$$b=\frac{1}{2x_{M}}\ln\left[-\frac{{F^{\prime }}_{A}\left({x^{\prime }}_{A}\right)}{{F^{\prime }}_{B}\left({x^{\prime }}_{B}\right)}\right]$$

The second transformation is carried out on the coordinates obtained after the first transformation, and the transformed data are recorded as $\left(x_{i}^{\prime \prime },F_{i}^{\prime \prime }\right)$. Thus, $x_{i}^{\prime \prime }=x_{i}^{\prime },F_{i}^{\prime \prime }=F_{i}^{\prime }e^{-bx_{i}^{\prime }}$ are substituted into Equation (13), and the following equation is obtained:(14)$$F^{\prime \prime }\left(x^{\prime \prime }\right)=F_{d}\left(x^{\prime \prime }\right)$$

The hysteresis loop after the secondary transformation is shown in Figure 1c; the curve is transformed into a hysteresis loop symmetric about the origin.

### 3.3. Identification of the Parameters k_1_, k_3_, c, and α

Bai et al. [22,23] concluded that the first-order linear stiffness and the third-order nonlinear stiffness play a decisive role in the distribution of the elastic restoring force. Therefore, the nonlinear elastic restoring force can be expressed as:(15)$$F_{k}^{\prime \prime }\left(x_{i}^{\prime \prime }\right)=k_{1}\left(x_{i}^{\prime \prime }\right)+k_{3}{\left(x_{i}^{\prime \prime }\right)}^{3}$$

Then, the nonlinear damping force is:(16)$$F_{c}^{\prime \prime }\left(x_{i}^{\prime \prime },{\dot{x}}_{i}^{\prime \prime }\right)=F\left(x_{i}^{\prime \prime }\right)-F_{k}^{\prime \prime }\left(x_{i}^{\prime \prime }\right)=F\left(x_{i}^{\prime \prime }\right)-k_{1}\left(x_{i}^{\prime \prime }\right)-k_{3}{\left(x_{i}^{\prime \prime }\right)}^{3}$$

The equivalent damping coefficient *c* and the damping component factor *α* can be obtained through the parameter identification of Equation (16), and the hysteresis loop of metal rubber after decomposition identification is shown in Figure 2.

## 4. Parameter Identification Results

### 4.1. Identification Results of the Parameters h and b and the Stiffness Coefficient

In order to obtain the accurate fitting function of the parameters, the pot-sharped metal rubber is prepared as shown in Figure 3 and the main characteristics is illustrated in Table 1. Herein, wires of austenitic 304 stainless steel (06Cr19Ni10) with diameters of 0.3 mm were selected as the raw material; this stainless steel is commonly used for the fabrication of mental rubber. Table 2 presents the chemical composition of 304 stainless steel. The preparation process was provided in Ref. [24]. The dynamic test data with a preload of 2.0 mm, a frequency of 2.0 Hz, and vibration amplitudes of 0.2, 0.4, 0.6, 0.8, and 1.0 mm are selected. The test data are calculated, and the evolution of the parameters *h* and *b* (Figure 4a) and the stiffness coefficients *K*_1_ and *K*_3_ (Figure 4b) with the vibration amplitude is obtained.

Polynomial fitting is selected for the parameter identification due to its simple structure and good fitting accuracy [24]. According to the curves of the parameters *h* and *b* as a function of amplitude shown in Figure 4a, the following equations are selected as the fitting functions:(17)$$h=\sum_{i}^{3}a_{i}A^{i}=a_{0}+a_{1}A+a_{2}A^{2}+a_{3}A^{3}$$
(18)$$b=\sum_{i}^{3}b_{i}A^{i}=b_{0}+b_{1}A+b_{2}A^{2}+b_{3}A^{3}$$

The fitting results are presented in Table 3.

As illustrated in Figure 4b, the stiffness coefficient gradually decreases with increasing amplitude, indicating that metal rubber presents soft characteristics [25]. The following equations are selected to fit *k*_1_ and *k*_3_:(19)$$k_{1}=\sum_{i}^{3}d_{i}A^{i}=d_{0}+d_{1}A+d_{2}A^{2}+d_{3}A^{3}$$
(20)$$k_{3}=\sum_{i}^{3}e_{i}A^{i}=e_{0}+e_{1}A+e_{2}A^{2}+e_{3}A^{3}$$

The fitting results are shown in Table 4.

### 4.2. Identification Results of the Parameters c and α

#### 4.2.1. Identification Results of the Parameter *c*

The hysteresis loop is divided into upper and lower branches; *c*_1_ is obtained from the test data of the upper branch of the hysteresis loop, while *c*_2_ is obtained from the lower branch of the hysteresis loop. The space surface of the parameters *c*_1_ and *c*_2_ is displayed in Figure 5.

As illustrated in Figure 5, the damping coefficient *c*_1_ decreases gradually with the increase of the loading amplitude, while the damping coefficient *c*_2_ shows an increasing trend. In addition, the damping coefficient maintains a relatively stable state with the increase of the frequency. Due to the variable damping and stiffness characteristics of metal rubber, the damping coefficient *c* and the damping component factor *a* of metal rubber have complex variation characteristics. The polynomial fitting has the characteristics of simple structure and high fitting accuracy [26]. Therefore, the polynomial was used to fit the damping coefficient and damping component factors. Moreover, the parameter fitting of metal rubber is generally 3 or 4 order [18,27]. After performing multiple fittings, it was found that the damping coefficient can be well fitted by the following equations:(21)$$c_{1}=\frac{p_{1}+p_{2}A+p_{3}\times A^{2}+p_{4}f+p_{5}f^{2}+p_{6}f^{3}}{1+p_{7}A+p_{8}A^{2}+p_{9}f+p_{10}f^{2}+p_{11}f^{3}}$$
(22)$$c_{2}=\frac{q_{1}+q_{2}A+q_{3}A^{2}+q_{4}A+q_{5}f+q_{6}f^{2}}{1+q_{7}A+q_{8}A^{2}+q_{9}f+q_{10}f^{2}+q_{11}f^{3}}$$
which are useful to better predict the damping performance of pot-shaped metal rubber. The fitting results are listed in Table 5 and Table 6.

#### 4.2.2. Identification Results of the Parameter *a*

The identification method of the damping component factor is similar to that of the damping coefficient; *a*_1_ is obtained from the test data of the upper branch of the hysteresis loop, while *a*_2_ is obtained from the lower branch of the hysteresis loop. The space surface of the parameters *a*_1_ and *a*_2_ is displayed in Figure 6.

It can be seen from Figure 6, the change in the damping component factors with amplitude and frequency is also complex. It can be seen from the figure that the damping component factors *a*_1_ and *a*_2_ decrease gradually with the increase of the loading amplitude. In addition, when the external load displacement is large (0.8 mm and 1.0 mm), the damping component factor becomes unstable with the change of frequency. After performing multiple fittings, it was found that the damping component factors *a*_1_ and *a*_2_ can be well fitted by the following equations:(23)$$\begin{array}{l} \alpha_{1}=m_{1}-m_{2}\times A-m_{3}\times f+m_{4}\times A^{2}-m_{5}\times A\times f+m_{6}\times f^{2}-m_{7}\times A^{3}+m_{8}\times A^{2}\times f-m_{9}\times \\ A\times f^{2}-m_{10}\times f^{3}+m_{11}\times A^{4}-m_{12}\times A^{3}\times f-m_{13}\times A^{2}\times f^{2}+m_{14}\times A\times f^{3}+m_{15}\times f^{4} \end{array}$$
(24)$$\begin{array}{l} \alpha_{2}=n_{1}-n_{2}\times A+n_{3}\times f+n_{4}\times A^{2}+n_{5}\times A\times f-n_{6}\times f^{2}+n_{7}\times A^{3}-n_{8}\times A^{2}\times f-n_{9}\times \\ A\times f^{2}+n_{10}\times f^{3}-n_{11}\times A^{4}+n_{12}\times A^{3}\times f+n_{13}\times A^{2}\times f^{2}+n_{14}\times A\times f^{3}-n_{15}\times f^{4} \end{array}$$

The fitting results are listed in Table 7 and Table 8.

The constitutive relationship model of the nonlinear function of the pot-shaped metal rubber can be obtained by substituting Equations (17)–(24) into Equation (16).

## 5. Experimental Verification of the Asymmetric Dynamic Model

According to the identification of the above parameters, the hysteresis loop of pot-shaped metal rubber can be reconstructed. In order to verify the accuracy of the established model, its results are compared with the experimental results, as shown in Figure 7. It can be seen from Figure 7a–d that the hysteresis of metal rubber obtained using the nonlinear dynamic model is in good agreement with the measured hysteresis loop. Therefore, the established model can well describe the changes in the displacement–force hysteresis curve of basin-shaped metal rubber and provide relevant guidance for engineering applications.

## 6. Vibration Performance Test

### 6.1. Response Calculation of the Pot-Shaped Metal Rubber Damper

The pot-shaped metal rubber damper is a type of passive vibration reduction, and its mechanical model is shown in Figure 8.

Assuming that the fundamental vibration of the damping system is a sinusoidal excitation, that is $u(t)=u_{0}\sin(\omega t)$, then the nonlinear constitutive relationship of the metal rubber isolator can be described as follows [18]:(25)$$g_{n}(y(t),\dot{y}(t),t)=F_{k}\left\{y(t)\right\}+F_{c}\left\{\dot{y}(t)\right\}=k_{1}y(t)+k_{3}y^{3}(t)+c{\left|\dot{y}(t)\right|}^{\alpha }\mathrm{sgn}(\dot{y}(t))$$
where $F_{k}\left\{y(t)\right\}$ is the nonlinear elastic restoring force, and $F_{c}\left\{\dot{y}(t)\right\}$ is a nonlinear damping force. Then, the differential equation of motion of the passive vibration isolation system can be written as:(26)$$m\ddot{x}+c{\left|\dot{x}-\dot{u}\right|}^{\alpha }\mathrm{sgn}(\dot{x}-\dot{u})+k_{1}(x-u)+k_{3}{\left(x-u\right)}^{3}=0$$

Introducing the relative displacement as:(27)$$y=x-u=y_{m}\sin(\omega t-\varphi )$$

According to a previous work [1], the equivalent damping coefficient can be expressed as:(28)$$c_{eq}=\frac{4}{\pi }\cdot (1-\frac{\alpha }{6})\cdot c\cdot {{(y}_{m}\cdot \omega )}^{\alpha -1}$$

By performing a trigonometric function transformation and using the harmonic balance method, the following higher-order transcendental algebraic equation can be obtained:(29)$${\left(-m\omega^{2}y_{m}+k_{1}y_{m}+\frac{3}{4}k_{3}y_{m}{}^{3}\right)}^{2}+{\left(\frac{\pi }{4}\left(1-\frac{\alpha }{6}\right){\left(y_{m}\cdot \omega \right)}^{\alpha }c\right)}^{2}={\left(mu_{0}\omega^{2}\right)}^{2}$$

The lag phase angle can be expressed as:(30)$$\varphi =\arctan\left[\frac{\frac{4}{\pi }\left(1-\frac{\alpha }{6}\right){\left(y_{m}\cdot \omega \right)}^{\alpha }c}{-m\omega^{2}y_{m}+k_{1}y_{m}+\frac{3}{4}k_{3}y_{m}{}^{3}}\right]$$

According to Equation (27), the output displacement of the system is:(31)$$x=y+u=y_{m}\sin(\omega t-\varphi )+u_{0}\sin(\omega t)=x_{m}\sin(\omega t-\tilde{\varphi })$$

Then:(32)$$\left\{\begin{array}{l} \tilde{\varphi }=\arctan\left[\frac{y_{m}\sin\varphi }{y_{m}\cos\varphi +u_{0}}\right]\\ x_{m}=\sqrt{y^{2}{}_{m}+u^{2}{}_{0}+2y_{m}u_{0}\cos\varphi } \end{array}\right.$$

By substituting $u_{0}$, *y_m_*, and φ into Equation (26), one can obtain $x_{m}$ and $\tilde{\varphi }$, and the displacement transmission rate of the negative vibration damping system then becomes:(33)$$T_{d}=\frac{x_{m}}{u_{0}}$$

### 6.2. Verification of the Sinusoidal Sweep Test

The sinusoidal sweep test is a vibration test in which the vibration frequency changes within a certain range while other vibration parameters remain unchanged. It is mainly used to determine the natural frequency and transmission rate of the damping system. The electromagnetic vibration test system (DC-4000, Suzhou Test Instruments Co., Ltd., Suzhou, China) was adopted in this study, which exhibits a low frequency distortion and a strong bearing capacity. The vibration test system is shown in Figure 9. The counterweight is 10 kg, the vibration magnitude is 3 g, and the sweep frequency is 5–200 Hz.

In order to verify the correctness of the nonlinear dynamic model and the parameter identification method, the displacement transmission rate curve of the negative damping system with the obtained parameters is calculated using Equations (29) and (30). The curve of the acceleration transmission rate as a function of frequency is obtained by performing a quadratic differentiation of the displacement transmission rate curve. Furthermore, the theoretical and experimental curves with the obtained parameters are compared, as shown in Figure 10. It can be seen from the figure that the theoretical curve is in good agreement with the experimental curve, which verifies the correctness of the dynamic model of the nonlinear vibration isolation system and the parameter identification algorithm.

## 7. Conclusions

In view of the current situation that most of the existing metal rubber is based on regular shapes, such as cylindrical, annular, and square, the pot-shaped metal rubber was proposed in this study. The asymmetric characteristics of pot-shaped metal rubber were explored, and the dynamic model and vibration performance of pot-shaped metal rubber components were investigated. The main conclusions are as follows:(1)Based on the parameter-separation concept, the hysteresis loop obtained from the dynamic test is accurately decomposed, and the nonlinear dynamic model of pot-shaped metal rubber is established. The specific expressions of the parameters of the model are obtained through parameter identification, and the accuracy of the model is verified.(2)The pot-shaped metal rubber damping system is analyzed theoretically. The experimental and theoretical acceleration transfer rate curves and the experimental and theoretical time-domain signals of the system under specific loading conditions, are compared. The results show that the theoretical curve is in good agreement with the experimental curve, which verifies the accuracy of the established model and the parameter identification method.

## Figures and Tables

**Figure 1 materials-15-05878-f001:**
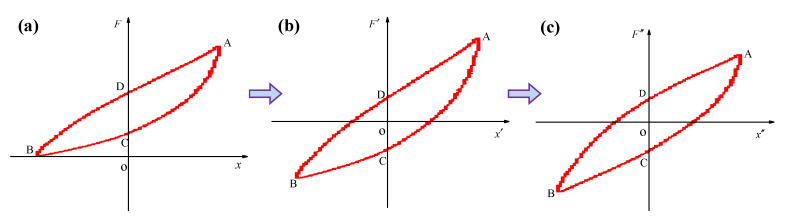
Hysteresis loops of pot-shaped metal rubber: (**a**) asymmetric hysteresis loop; (**b**) after the first transformation; (**c**) after the second transformation.

**Figure 2 materials-15-05878-f002:**
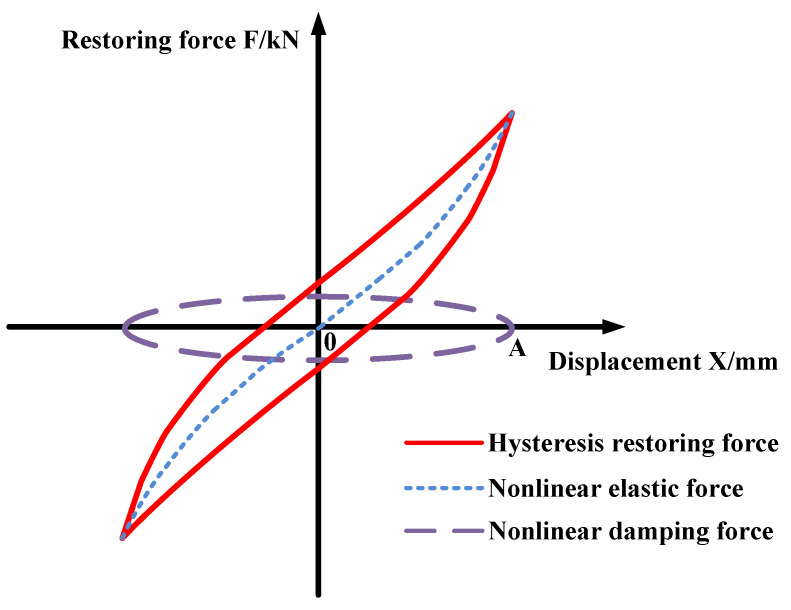
Exact decomposition of the hysteresis loop of metal rubber.

**Figure 3 materials-15-05878-f003:**
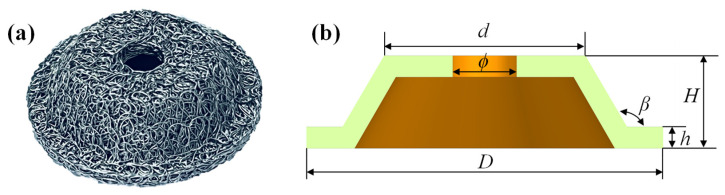
(**a**) The fabricated pot-shaped metal rubber, and (**b**) the dimensions of metal rubber.

**Figure 4 materials-15-05878-f004:**
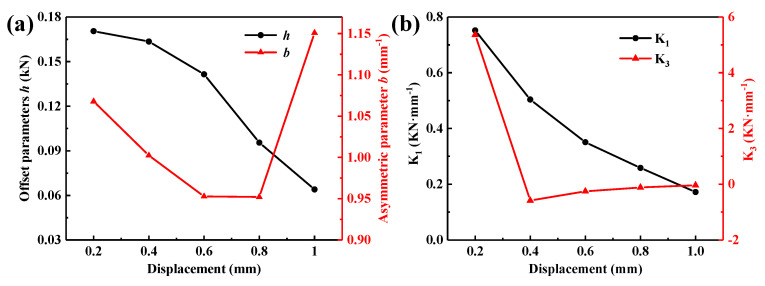
(**a**) Evolution of the parameters *h* and *b* with amplitude and (**b**) evolution of the stiffness coefficient with amplitude.

**Figure 5 materials-15-05878-f005:**
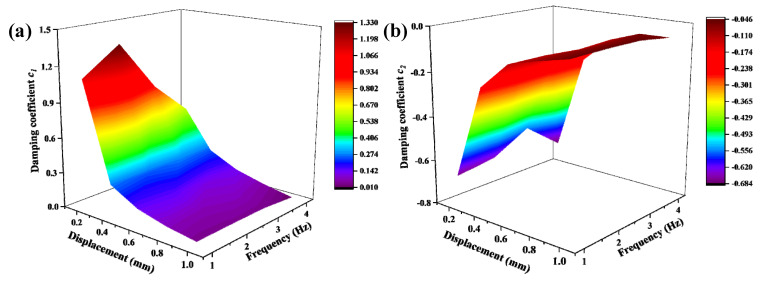
Variation trend of damping coefficient with amplitude and frequency (**a**) *c*_1_ and (**b**) *c*_2_.

**Figure 6 materials-15-05878-f006:**
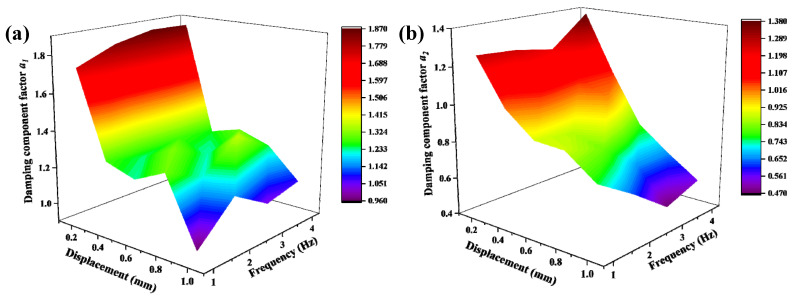
Variation trend of damping component factors with amplitude and frequency (**a**) *a*_1_ and (**b**) *a*_2_.

**Figure 7 materials-15-05878-f007:**
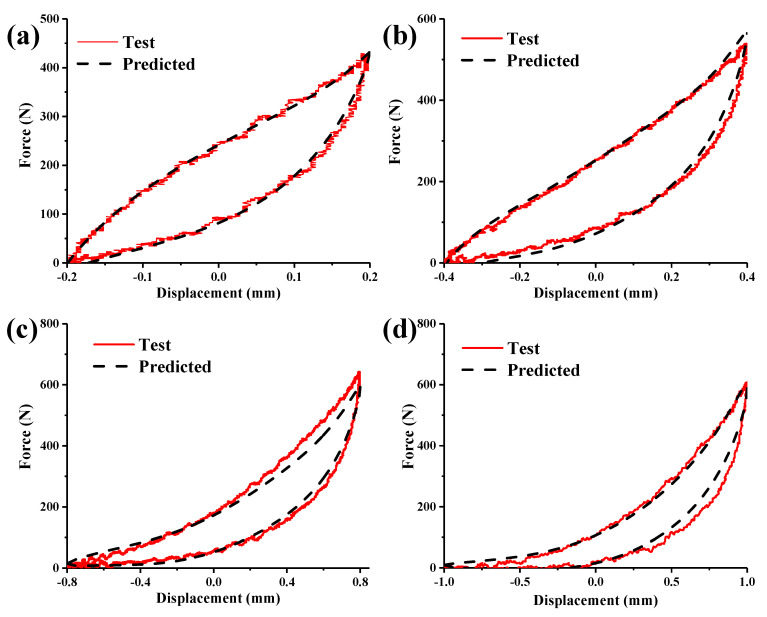
Comparison of the predicted force–displacement curves of pot-shaped metal rubber with the experimental results: (**a**) 0.2 mm, (**b**) 0.4 mm, (**c**) 0.8 mm, and (**d**) 1.0 mm.

**Figure 8 materials-15-05878-f008:**
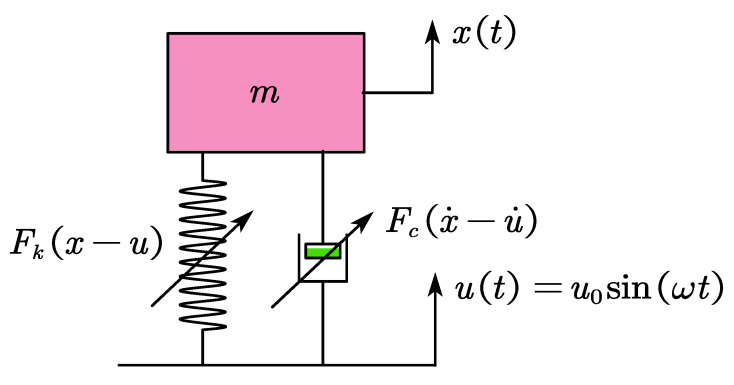
Passive vibration reduction model of pot-shaped metal rubber.

**Figure 9 materials-15-05878-f009:**
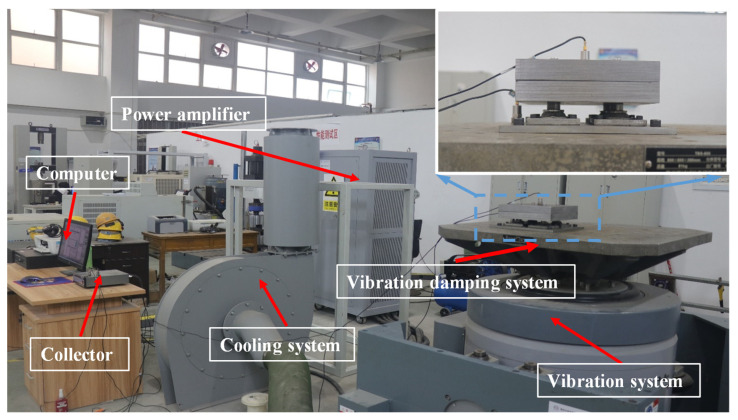
Vibration test system.

**Figure 10 materials-15-05878-f010:**
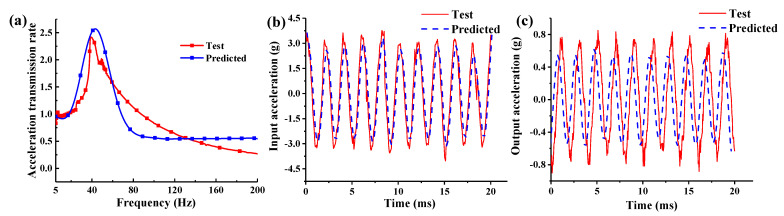
Comparison between the experimental and predicted curves: (**a**) acceleration transmission rate, (**b**) input acceleration, and (**c**) output acceleration.

**Table 1 materials-15-05878-t001:** Main characteristics of pot-shaped metal rubber.

*β*/^o^	*D*/mm	*d*/mm	*ϕ*/mm	*H*/mm	*h*/mm
120	50	28.4	9	13	3

**Table 2 materials-15-05878-t002:** Chemical composition of the 06Cr19Ni10 material.

Element	C	Si	Mn	P	S	Ni	Cr	Fe
Wt.%	0.06	0.86	1.84	0.02	0.004	8.23	18.52	Bal.

**Table 3 materials-15-05878-t003:** Fitting results of the parameters *h* and *b*.

Parameter	*a* _0_	*a* _1_	*a* _2_	*a* _3_	*b* _0_	*b* _1_	*b* _1_	*b* _3_
Value	0.134	0.306	0.685	0.307	1.022	0.626	2.410	1.911

**Table 4 materials-15-05878-t004:** Fitting results of the stiffness coefficients.

Parameter	*d* _0_	*d* _1_	*d* _2_	*d* _3_	*e* _0_	*e* _1_	*e* _2_	*e* _3_
Value	1.147	−2.412	2.374	−0.938	18.780	92.920	140.100	−66.050

**Table 5 materials-15-05878-t005:** Fitting results of the damping coefficient *c*_1_.

Parameter	*p* _1_	*p* _2_	*p* _3_	*p* _4_	*p* _5_	*p* _6_
Value	0.043	−0.111	−0.103	−0.055	0.116	−0.063
Parameter	*p* _7_	*p* _8_	*p* _9_	*p* _10_	*p* _11_	
Value	1.138	0.306	−0.054	0.173	−0.124	

**Table 6 materials-15-05878-t006:** Fitting results of the damping coefficient *c*_2_.

Parameter	*q* _1_	*q* _2_	*q* _3_	*q* _4_	*q* _5_	*q* _6_
Value	−0.052	0.034	0.136	0.063	−0.005	0.004
Parameter	*q* _7_	*q* _8_	*q* _9_	*q* _10_	*q* _11_	
Value	1.556	0.589	−0.019	0.057	−0.032	

**Table 7 materials-15-05878-t007:** Fitting results of the damping coefficient *a*_1_.

Parameter	*m* _1_	*m* _2_	*m* _3_	*m* _4_	*m* _5_	*m* _6_	*m* _7_	*m* _8_
Value	83.298	−13.673	−166.229	29.842	−0.697	116.474	−27.175	1.550
Parameter	*m* _9_	*m* _10_	*m* _11_	*m* _12_	*m* _13_	*m* _14_	*m* _15_	
Value	−0.162	−33.266	8.024	−0.174	−0.207	0.052	3.324	

**Table 8 materials-15-05878-t008:** Fitting results of the damping coefficient *a*_2_.

Parameter	*n* _1_	*n* _2_	*n* _3_	*n* _4_	*n* _5_	*n* _6_	*m* _7_	*m* _8_
Value	−0.625	−3.946	4.901	0.970	3.577	−3.737	4.923	−3.902
Parameter	*n* _9_	*n* _10_	*n* _11_	*n* _12_	*n* _13_	*n* _14_	*n* _15_	
Value	−0.628	1.091	−2.962	0.688	0.537	0.001	−0.108	

## Data Availability

The data presented in this study are available on request from the corresponding author. The data are not publicly available due to privacy.

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
