# Peer review of "Study of the Dynamic Model and Vibration Performance of Pot-Shaped Metal Rubber"

_materials, 2022, doi:10.3390/ma15175878_

Round 1

Reviewer 1 Report

Herein,

The authors investigate theoretical calculations on the damping ability of pot shaped metal rubber. The manuscript is well-written and the content of the figures is sufficiently clear.

I recommend that the authors discuss why the pot shaped and what chemicals/composition are the most prevalent metal rubbers comprised of.

Author Response

Dear Reviewers: Thank you for your letter and for the reviewers’ comments concerning our manuscript entitled “Study of the dynamic model and vibration performance of pot-shaped metal rubber”. Those comments are all valuable and very helpful for revising and improving our paper, as well as the important guiding significance to our researches. We have studied comments carefully and have made correction, which we hope meet with approval.

In the present document, text and other features corrected or added to the revised manuscript are in red mark.

Reviewer 2 Report

In this manuscript Yiwan Wu et al. explain the manuscript “Study of the dynamic model and vibration performance of pot-shaped metal rubber”. This manuscript is interesting, but many details are intolerable. Before acceptance, the following remarks should be addressed:

1.      Select type of the paper.

2.      Authors’ mails are not reflected in the author list.

3.      References should be corrected to the standard style of Materials. Please check all references. In addition, the reference 23 should be easily accessible, please correct.

4.      It is not possible to accept texts with the legend: Error! Reference source not found…

5.      Correct the Table 2, 4 and 5.

6.      Explain extensively where alpha 1 and alpha 2 come from (equations 17 and 18).

7.      The authors must emphasize the importance of the metal rubber complex in the conclusions, for example, the applications and the important findings of the materials in comparation with other authors.

Author Response

Dear Reviewer:

Thank you for your letter and your comments on our manuscript entitled "Study of the dynamic model and vibration performance of pot-shaped metal rubber". These comments are very helpful to the revision and improvement of our paper, and have important guiding significance for our research. We have carefully studied the comments and made corrections in the hope of approval.

Reviewer 3 Report

The authors model the nonlinear dynamics of pot-shaped metal rubber based on the analysis and study of the asymmetric model of the wire rope shock absorber combined with the trace model. The parameters in the model are identified based on the parameter-separation identification method and verified experimentally. Even though the manuscript seems to be rigorously argued from a mathematical point of view, overall the paper is not well written. It is very hard to follow and understand.

The reviewer would like to pose the following questions related to the manuscript:

 1.      The authors assume everyone is familiar with the Bouc–Wen hysteresis model. Provide a reference for those who may not be familiar with it. The modified equations 2 (a) –(d) are functions of x, while equations 1 (a) –(d) are functions of time. There is no explanation given for the modification. Reference is also not provided for equations 3 (a)- (d) yet they have not been derived. Some parameters in equations are not defined. Do not assume that everyone knows what they are.

2.      It is not clear if figure 2 is a schematic illustration or based on any particular equations.

3.      Some figures and figure captions do not explain the details that the authors wish to express, for example, figure 3 and figure 5.  

4.      It is not clear how equations 15 and 16 are arrived at. This statement “After performing multiple fittings, it was found that the damping coefficient can be well fitted by the following equations” is confusing. Provide the most general form of equations and show plots of how the selected parameters provide the best fit. The same goes with equations 17 and 18, it is not clear how these particular parameters were arrived at.

5.      How unique are the fitting parameters given in tables 5, 6, 7, and 8? How was the fitting data optioned? Experimental details are not provided.

6.      Provide a reference for equation 19

7.      Results are not properly discussed. Stating that “the change in the damping coefficient component factor with amplitude and frequency is more complex than the change in the stiffness coefficient” does not explain anything.

Here are some suggestions to help improve the readability of the paper.

1.      Be sure to use subscripts for terms in line 26 on page 3.

2.      Section 3.2.2 line 126 page 8 should be alpha not “a”

Author Response

Dear Editors and Reviewers:

Thank you for your letter and for the reviewers’ comments concerning our manuscript entitled “Study of the dynamic model and vibration performance of pot-shaped metal rubber”. Those comments are all valuable and very helpful for revising and improving our paper, as well as the important guiding significance to our researches. We have studied comments carefully and have made correction, which we hope meet with approval.

Round 2

Reviewer 2 Report

I appreciate you have heeding my recommendations. The manuscript can be published.

Author Response

Thank you for your letter and for the reviewers’ comments concerning our manuscript entitled “Study of the dynamic model and vibration performance of pot-shaped metal rubber”. Those comments are all valuable and very helpful for revising and improving our paper, as well as the important guiding significance to our researches. 

Reviewer 3 Report

The authors have tried to address my original concerns.